# Being at the Bottom Rung of the Ladder in an Unequal Society: A Qualitative Analysis of Stories of People without a Home

**DOI:** 10.3390/ijerph16234620

**Published:** 2019-11-21

**Authors:** Mzwandile A. Mabhala, Asmait Yohannes

**Affiliations:** 1Department of Public Health and Wellbeing, Faculty of Health and Social Care, University of Chester, Riverside Campus, Chester CH1 1SL, UK; 2Asmait Skin Care, Southfield Ave, Stamford, CT 06902, USA; info@asmaitskincare.com

**Keywords:** homeless(ness), social justice, social status, social mobility, social engagement

## Abstract

*Background:* Homelessness is rising in the United Kingdom, despite investment in measures to eradicate it made by the government and charity organisations. *Aim:* The aim is to examine the stories of homeless people in order to document their perceptions of their social status, the reasons that led to their homelessness, and propose a conceptual explanation. *Method*: We conducted 26 semi-structured interviews in three centres for homeless people in Cheshire, North West of England. *Results:* Three categories—education, employment, and health—emerged from the data and provided a theoretical explanation for the reasons that led to their homelessness. These are vital not only for the successful negotiation of one’s way out of homelessness, but also for achieving other social goods, including social connections, social mobility, and engaging in positive social relationships. *Conclusion*: Participants catalogued the adverse childhood experiences, which they believe limited their capacity to meaningfully engage with the social institution for social goods, such as education, social services, and institutions of employment. Since not all people who have misfortunes of poor education, poor health, and loss of job end up being homeless, we contend that a combination of these with multiple adverse childhood experiences may have weakened their resilience to traumatic life changes, such as loss of job and poor health.

## 1. Introduction

Homelessness is rising in the United Kingdom (UK) [1,2]. In January 2018, the number of rough sleepers in England was estimated to be 4751; since 2010, an additional 2983 people have been estimated as sleeping rough in England on any given night [2]. This represents an increase of 169% in the last seven years [2].

The UK government and charity organisations are making efforts to mitigate the effects of homelessness [1,2]. In 2017, the UK government published the “Homelessness Reduction Act”, which provided local authorities with guidelines on reducing homelessness [3]. The guidance on implementation was published in April 2018. The UK government is also investing £1 billion annually until 2020 to reducing homelessness and rough sleeping, including £28 million on piloting the Housing First approach in Manchester, Liverpool, and West Midlands, £20 million to support homeless people and those at risk of homelessness, and £9 billion by 2021 to build new homes [2]. However, all the efforts remain solely focused on providing accommodation or preventing the loss of it.

The current understanding is that the causes of homelessness are more complex than the absence of a place to live [4,5,6]. Several researchers categorise them into individual factors and structural factors [7,8,9,10,11,12]. Individual factors are broadly associated with the circumstances or behaviours that could increase a person’s vulnerability to becoming homeless. Some the examples of individual factors that emerged from the literature include adverse childhood experiences (ACEs), mental health and substance misuse problems, personal history of violence, and criminal justice system association [9,11]. 

Structural factors are associated with the failures of social policy, society, and social institutions to prevent homelessness [9,11]. Some of the examples include the absence of low-cost housing, employment opportunities for low-skilled workers, income support, poor education, low education participation, and achievement [9,13]. Furthermore, studies posit that the failure of social policies and social institutions to tackle socioeconomic discrimination against ethnic minorities, lesbian, gay, bisexual, transgender, queer, and two-spirit (LGBTQ2S) and those with mental illnesses or cognitive disabilities increase their vulnerability to homelessness [13,14,15].

Increasingly, researchers are investigating how the interaction between the structure and individual factors cause homelessness [11,16,17]. The consensus is that poverty plays a pivotal role in many pathways to homelessness [18,19,20]. 

Researchers are investigating and reporting how poverty act as a catalyst in pathways to homelessness. Deforge et al. found that poverty marked by residential instability was a precursor to adverse life events [17]. In their study, they found that for participants with a history of homelessness, insufficient income was the primary cause of their homelessness. They reported that insufficient income was a catalyst to other factors, including family or domestic instability, lack of employment, lack of suitable housing, addictions to alcohol or drugs, lack of support from friends and or family, physical ill-health, mental health condition, and a prison or jail record [17]. These findings are consistent with several studies that reported that being poor and homeless are predictive of adopting maladaptive behaviour, such as engaging in criminal activities, trading sex for money, and selling or using drugs [14,21]. 

A systematic review by Fry et al. demonstrated that young people who had experienced homelessness were more likely from poor households or foster care and experienced abuse or neglect; in turn, these young people were more likely to demonstrate poorer cognitive functioning than those who had not had these experiences [16]. Similarly, several studies reported that the prevalence of ACEs’ predictors of homelessness, such as being raised in a care home or foster care, disrupted domestic education violence, housing instability and physical, emotional, and sexual abuse at an early age, are higher amongst the poor [11,17]. Shinn et al. indicated that the children in low-income households are more likely to experience two or more ACEs compared to children in medium to high income households [22]. 

Furthermore, the evidence suggests that some clusters of ACEs are more predictive of homelessness than others [17,22]: a cluster of childhood problems, including mental health and behavioural disorders, poor school performance, a history of foster care, and disrupted family structure, was most associated with adult criminal activities, adult substance use, unemployment, and subsequent homelessness [22,23,24,25,26,27,28]. Barker et al. indicated that four forms of ACEs (physical abuse, emotional abuse, physical neglect, and emotional neglect) were significantly and independently related to not completing a high school education, particularly physical abuse [28].

The multifaceted nature of causes of homelessness shapes the debate about whether the policy on homelessness should focus on tackling poverty as a fundamental cause of homelessness or on immediate causes and harmful effect of homelessness. We envisage that this study will contribute to the understanding of the pathways to homelessness, and, thus, reorient the approaches to tackling homelessness from the current focus on providing accommodation to addressing the fundamental determinants.

Despite the evidence, to the best of our knowledge, no study was found that examined the participants’ life stories to provide a theoretical explanation on the potential pathways by which social conditions, ACEs, and maladaptive behaviour cause homelessness. This study aimed to examine the stories of homeless people in order to document their perceptions of their social status as homeless people, the reasons that led to their homelessness, and propose a conceptual explanation. It is the second article from the project that investigated the determinants of homelessness. Mabhala et al. [6] presented a theoretical explanation of the social conditions within which becoming homeless occur. The main contribution of this paper is that for the first time, as far as the researchers’ knowledge is concerned, it offers a model to examine the potential pathways by which socioeconomic and political conditions, individual circumstances, and maladaptive behaviour cause homelessness.

## 2. Study Methods

Constructivists grounded theory (CGT) was considered to be epistemologically and ontologically consistent with the aims and the context of this study [29]. The first aspect of CGT that epistemologically and ontologically fit with the aim of this research relate to the researchers’ evidence-based belief that the phenomenon of homeless is situated within a broader context of socioeconomic determinants of inequalities and social justice [6]. This is consistent with a constructivist grounded theory that recognises the importance of social context within which the data collection and analysis is situated [29]. 

The second is CGT’s recognition that the researcher brings to the research field with their values, beliefs, and experiences, which shape the conduct of research and interpretation of data [29]. In several publications, Mabhala [6,30,31,32] argued that social inequalities are created by social policies and institutions of society favouring certain starting places for some over others [6,33]. He argued that social justice principles are the foundation for the strategies to reduce the socioeconomic inequalities [6,31]. The researcher’s established position is consistent with the constructivist grounded theory’s view that differences in power and opportunities maintain and perpetuate differences in social inequalities [29,30,31,34,35]. The CGT approach advocates for the investigation of conditions under which such differences arise and are maintained [14] (p. 131). 

The third is CGT’s acknowledgement that the researcher’s resulting theoretical explanation constitutes the researcher’s interpretation of the meanings that the participants ascribe to their situations and actions in their contexts [29]. This outlines the CGT’s fundamental ontological belief in multiple realities constructed through the experiences and understandings of different participants’ perspectives and is generated from their different demographic, social, cultural, and political backgrounds [29].

The fourth is CGT’s epistemological belief that knowledge is shaped by the cultural, historical, and social norms that operate within that context and time. These assumptions outline the importance of taking account of the influence of the researcher’s involvement, and the influence of the contexts that surround data collection, analysis, and interpretation of findings [29]. These assumptions support the researchers’ desire to learn how and to what extent the homelessness is embedded in wider social contexts.

### 2.1. Setting, Recruitment, and Sampling Strategies

The research took place in three centres for homeless people in two cities (Chester and Crewe) in Cheshire, UK. Chester is the most affluent city in Cheshire, and Crewe is the least affluent. One of the main considerations for the recruitment strategy was to ensure that the process complied with the ethical principles of voluntary participation and the equal opportunity to participate. To achieve this, an email was sent to all the known homeless centres in the Cheshire and Merseyside region, inviting them to participate. Three homeless centres agreed to participate, all of them in Cheshire—two in Chester and one in Crewe. 

Before commencing the research study, M.M. spent one day a week for three months in three participating centres. During that time, oral presentations of the study were given to all users of the centres, inviting all the participants to participate, and a written participants information sheet was provided to those who expressed wishes to participate.

To help potential participants make a self-assessment of their eligibility to participate without unfairly depriving others of the opportunity, the participant information sheet outlined the criteria that potential participants had to meet. The criteria were as follows: all participants must be people who have no permanent home, are dependent on homeless people’s facilities for a place to stay, safety, food and healthcare, must be able to speak English, and, at the time of interview, must be free from physical pain or discomfort [20,21]. All the willing participants gave their names to the site managers, and the site managers assessed and confirmed their eligibility. All those who volunteered met the participation criteria. All participants who volunteered and met the criteria had recent experience of rough sleeping, and they were predominantly men (see Table 1). 

We used two sampling strategies: purposive and theoretical. We started with purposive sampling and conducted in-depth one-to-one semi-structured interviews with eight homeless people to generate themes for further exploration. As categories emerged from the data analysis, we adjusted the interview guide to allow an in-depth examination of the emerging categories. We used theoretical sampling to conduct a further eighteen interviews to refine undeveloped categories in accordance with Strauss and Corbin’s [36] recommendations. 

We carried out a total of 26 semi-structured interviews. The sample comprised of 22 males and 4 females; the youngest participant was 18, the eldest was 74 years, and the mean age was 38.6 years. Table 1 illustrates participant’s education history, childhood living arrangements, brief participants’ family and social history, emotional and physical health, and the onset of and trigger for homelessness.

### 2.2. Data Collection

Semi-structured interviews were used to collect data. Each interview was 45 to 60 minutes in length. All interviews were conducted within the centres of homeless people’s offices.

Interviews were tape-recorded and then transcribed verbatim by (A.Y.). Data collection continued until ‘theoretical sufficiency’ [37] (p. 117) was achieved. We adopted the same meaning of theoretical sufficiency as Díaz Andrade [37], which is ‘that categories have been developed to a sufficient extent so that it is possible to explore their relationships and draw some conclusions’ [37] (p. 47). Some would describe this as theoretical saturation [37]; however, in this study, the term ‘theoretical sufficiency’ was considered more appropriate. While both indicate that the data have been properly analysed, the latter acknowledges that the process of generating categories can never be absolutely exhaustive [37,38]. 

### 2.3. Ethical Approval 

The Faculty of Health and Social Care Research Ethics Subcommittee of the University of Chester granted ethical approval for this study. The committee was satisfied that it complied with the Economic and Social Research Council’s Framework for Research Ethics (ethic code: RESC1115-666) [39]. Consistent with this framework, at the time of consenting to and commencing the interview, the participants were required to appear to be under no influence of alcohol or drugs, have a capacity to consent as stipulated in the England and Wales Mental Capacity Act 2005 [40], be able to speak English, and be free from physical pain or discomfort.

To preserve confidentiality and anonymity of participants, we did not include participants’ names on the audiotape—the recording only commenced after the introduction and small talk had been completed. 

During the interview process, participants were not to be addressed by name, and pseudonyms were used to identify transcripts during the analysis and dissemination. Electronic data were stored on a password-protected computer. All study data will be kept for a minimum of ten (10) years from the date of final publication of the findings 

To ensure informed consent, the investigator read out the information sheet for each participant, then provide them with a hard copy to keep. Participants were given seven days to consider whether they wished to participate or not.

## 3. Data Analysis

Consistent with the requirements of grounded theory, we collected and analysed data simultaneously. The analysis drew on the grounded theory principles of constant comparative analysis and the iterative process of data collection and data analysis to build the theory inductively [36]. We broadly organised data around two phases of comparative analysis—making a constant comparative analysis and making a theoretical comparison [41]—a process summarised in Figure 1.

The key question that we asked all participants in this study was: could you tell us about your life story from where and when you were born to the point you became homeless? 

Our analysis departed from the premise that people say something because it means something to their experience. Therefore, our task was to discover the meanings ascribed to the description of their experiences. At the comparative analysis phase (Figure 2), we read each interview line-by-line to identify segments of data that contained theoretically significant incidents: incidents in the data that appeared to have the potential to render a theoretical explanation to what the participants ascribe to being homeless.

All participants attributed being homeless to their adverse childhood experiences (ACEs). The typical response to the above question was—*“I did not go to school…”, “I left school when I was fifteen, etc.”, “I was bullied at school”,* “*I had a childhood of so much persistent and consistent abuse from my mother and my stepfather”, “I was neglected by my mum”, “He abused me when I was five and raped me when I was six”.*

We listed all these in-vivo codes: the data labels that come directly from actual spoken words of the participants. We gradually shortened them into concepts that captured the reasons participants ascribed to being homeless [36]. We grouped similar concepts to develop categories [41]. 

After a period of interrogation of data, three categories (Figure 2)—education, employment, and health—emerged from the data and provided a theoretical explanation for how the participants conceptualised their homelessness status. 

Once the categories had been identified, we began to group the codes around the category they represented (Figure 2). For example, we interrogated the data to discover what participants meant by education in relation to homelessness. Participants’ statements such as: *“If you don’t have education early enough in life, I don’t mean maths, English and that, I mean being able to actually engage with people”* sensitised us to code all the incidents in data that provide a theoretical explanation to what participants ascribed to education. Figure 2 indicates three properties that emerged from the data that provided the participants’ experience and perceptions of education—social interaction, social mobility, and engagement in social relationships. 

Once we established the relationships between categories and associated properties (subcategories), we focused on relating categories to other categories (Figure 2). This stage of the analysis revealed participants’ perception of the interaction among three categories as well as the crosscutting interaction between categories and subcategories (Figure 2). The final stage of our analysis consisted of a combination of theoretical comparison and theoretical sampling (Figure 1). We theoretically compared the emerging categories with the existing literature [36]. In doing so, we filled in and refined the poorly defined categories. This process continued until theoretical sufficiency was achieved [37]. While we considered all individual participants stories, we selected a few extracts that best represented the meanings that participants ascribed to their experiences. 

## 4. Results:

### 4.1. Being Homeless is Being at the Bottom Rung of the Ladder 

As Figure 2 indicates, participants’ stories gave a sense that homelessness reflects an unequal society. They equated the low social status of being homeless with being at the bottom rung of the “social” ladder, with general society looking down upon them. Participants did not use the concept “unequal society”; however, their descriptions of their relationship with the general public were replete with references to being at the “bottom rung of the ladder”, being “labelled as robbers and users”, being "kicked about”, being “looked down upon”, being “discriminated” against, etc. This implies that they believe that society treats them differently due to their low social status as homeless people. 

The analysis revealed that the low social status ascribed to being homeless made them vulnerable to discrimination and physical violence by members of the public. All participants who experienced physical violence, injuries, and hospital admission asserted that members of the public assaulted them. As Kelly explained:


*I am sick of being looked down upon by the members of the public; I am sick of being discriminated and being hurt because I am homeless on the street. …you get assaulted left, right and centre by the members of the public and at the same time police are giving us all the ASBOs[anti-social behaviour order] kicking us out of town, shift you off to another town, what is the point of that? You will still be homeless.*
[Kelly]

According to Kelly and Thomas, homeless women and children are particularly more vulnerable than male counterparts:


*It is horrible being in the street you … you get battered every so often. … people come up to me on the street kick punch me, which is what they do all the time because I am a homeless person in the street.*



*I am a big girl … sometimes they think I am a man, but I do not think that matters anyway, they have known I am a girl, but they still battered me.*
[Kelly]

Thomas’s story illustrates the vulnerability of homeless children to physical assault and harm: he explained that he was only twelve years old when he was regularly experiencing physical violence by members of the public:


*I used to be battered by people in their night out in the town centre where I grew up, people used to batter me, I don’t think they understood I was that I was the only child, yea I used to be battered by drunk people Friday and Saturday night.*
[Thomas]

Clarke was only fifteen when he became homeless; his story illustrates the vulnerability of being a homeless child and in fear of being harmed:


*…after I got kicked out … I was living and sleeping on the streets, …going to work then going on the streets again… I tried to find any shelter I could go under where it isn’t wet, where it is out of the way somewhere where no one would disturb me… I was quite scared I suppose ’cause I didn’t know where to go or what was gonna happen with my job ’cause I was living on the streets.*
[Clarke]

For a homeless child, the threat is not only confined to physical health and safety, as illustrated by his statement, “to find any shelter I could go under where it isn’t wet, where it is out of the way somewhere where no one would disturb me”; there are also threats to social wellbeing, as illustrated by the importance he attached to keeping his employment. At the time of this interview, Clarke was 17 years old, homeless, and working in McDonald’s. His story further illustrates the challenges of being young and homeless:


*I’ve been trying to bid on properties… they wanted non-smokers or people over 21, stuff like that… [be]cause I’m 17, on a zero-hour contract they were like what if you don’t have the money to pay rent or…*
[Clarke]

Participants described a frightening feeling of loss of self-respect that comes with being subjected to assault and discrimination and the indignity of lack of privacy:


*You just seem to lose all self-respect and reassurance that you have when you’ve got a roof over your head that you can go back to, um frightening very frightening I … It is really horrible not being able to do anything in the privacy of your own home like you know. Frightening very frightening, constant fear of being robbed, bullied and turning to drink, or drugs.*
[Danny]

By way of example, Gary recalls a humiliating incident when a member of the public burnt his tent with all his belongs in it:


*When I got that injury and had to go to the hospital, um somebody had burnt me tent down with all my belongings in it, whilst I was at the hospital. So, I lost everything.*
[Gary]

Thomas and Marie reported a similarly frightening incident when a member of the public burnt their tent with the two of them inside it:


*We came to Chester because Chester has facilities like this that accommodate homeless people. But because of no local connection, there’s not a lot of people can do. So we put up a tent near the canal until this Saturday when people decided to trash it.*
[Marie and Thomas]

Participants reported constant fear not only of members of the public harming them and destroying their possessions but also of other homeless people robbing them. This is just one of many incidents:


*My stuff gets robbed off me all the time, my medication gets robbed. Last week all my medication and all that got robbed. Cause the diazes and zoppies and that people get off on them, gets them wrecked.*



*If I’ve got 14 Ensures in a bag, walking around with them all the time and a big bag full of medication I’m asking to get robbed, in the town I’m asking to get robbed. You can’t even leave your *dibs* on the floor sometimes and someone robs them off ya. You know Ensures in the shop they are £4.50 a bottle so people are gonna steal them off me cause they could get £1 each for them on the streets. But I can’t have people doing that all the time.*
[Henry]

### 4.2. Social Status of Homeless People

According to the participants, the low social status ascribed to homeless people emanated from their poor outcomes in social goods, such as education, employment, and health.

We labelled these social goods as “foundations for social status”. Participants themselves did not use this term to describe them; however, their descriptions clearly implied that they perceive them as fundamental factors that created and sustained their pathways to becoming homeless. They explained that these are vital not only for the successful negotiation of one’s way out of homelessness, but also for achieving other social goods, including social engagement, connections, social mobility, and engaging in positive social relationships. Figure 2 illustrates the inextricable interaction between education, employment, health, and being homeless: the word *interaction* is preferred over association or relationship because this study proposes a conceptual explanation rather than a cause and effect association.

#### 4.2.1. Education

Barry described the kind of education he is envisioning, something that would enable a young person to develop social skills such as the ability to engage in positive social interaction, make social connections, and develop full relationships with others. He explains that:


*… at the end of the day, if you don’t have education early enough in life, I don’t mean maths, English and that, I mean being able to actually engage with people. If you don’t have that education early on enough in life, I’m talking about the age of 5, 6, 7, 8, if you don’t have that then we get a despondent youth and once it goes down, they’re on a very slippery slope. And it’s very tough to get yourself back up, you know. You are on a ladder without rungs if you know what I mean.*
[Barry]

Figure 2 illustrates participants’ perceived role of education in their low social status, and Barry compares a homeless person without the kind of education he described with being at the bottom of “a ladder without rungs”. This implies that he perceives a homeless person without education as having limited opportunities for social mobility. 

Several participants expressed similar views. For example, when the interviewer asked Pat to describe how he became homeless, his response was:


*I did not go to school because I kept on bunking. When I was fifteen I left school because I was caught robbing, the police took me home and my mum told me you’re not going back to school again, you are now off for good. Because if you go back to school you keep on thieving, she said I keep away from the lads. I said fair enough. When I was seventeen I got run over by a car.*
[Pat]

Pat implies that had he been educated he would not have been in a better position in society. His second sentence “…I was fifteen when I left school…” is a variant on one of the two most frequently recurrent themes that emerged from participants’ stories as they describe education as a major contributor to their low social status, namely: Leaving school at the age of 16Poor educational experiences.

All but one of the participants left school at the age of sixteen or younger, and all constructed leaving school at sixteen as part of the reason for their low social status (see Table 1). They described the varying activities they engaged in following their exit from the educational system. For instance, Lee describes the activities that he engaged with since leaving school, which he believed set his path towards homelessness:


*I left school when I was fifteen because I was the youngest in my year. I left school then I went off the rails. I got kidnapped for three and a half months. When I came back I was just more interested in crime. When I left school I was supposed to go to College, but I went with travellers. I was just more interested in getting arrested every weekend until my mum say right I have enough of you. I was only seventeen. I went through the hostels when I was seventeen.*
[Lee]

Danny indicated that his path to homelessness began when he left school at sixteen and went straight into marriage:


*I left school when I was sixteen straight away I got married had children, I have three children and marriage was fine. Um, I was married for 17 years. Um, and as the marriage broke up I turned to…*
[Danny]

The context within which Danny described leaving school at age sixteen and getting married at that age implies that this combination was the catalyst for his path to homelessness. Similarly, Alvin describes a combination of leaving school at sixteen and going straight into working on construction sites: 


*… I was working on building sites before I left school at 16 you know when I was doing my exams I actually got the job and I was hod carrying for the bricklayers…*
[Alvin]

In all cases, when the route they took after leaving school came to a dead-end, their world fell apart, and they ended up homeless.

In term of school experiences, lack of support from the teachers and negative peer influences were some of the issues reported by participants as reasons for their poor educational outcomes. Marco’s description represents the extreme form of poor schooling experience: 


*On a few occasions I came out on the corridors, I would be getting battered on to my hands and knees and teachers walk past me. There was quite often blood on the floor from my nose, would be punched on my face and be thrown on the floor. …. It was a hard school, pernicious. I would go as far as saying I never felt welcome in that school, I felt like a fish out of the water.*



*Eventually, I started striking back when I started striking back suddenly I was a bad one. My mother decided to put me in … school for maladjusted boys, everyone who been there including me has spent time in prison.*
[Marco]

Several participants cited the experience of being bullied at school as a cause of leaving at sixteen or younger. Like most of the participants in this study, Kelly was bullied by other children because of her adverse family circumstances:

*The kids used to overhear my aunty talking to the teachers about my child abuse* [she was sexually abused by her biological father]*, I used to go to the toilet and the boys will say are you coming out to play? The girls will be telling me we don’t want anything to do with because of your daddy something, something, something.*[Kelly]

Marie and Thomas come from similar family backgrounds and expressed similar experiences. Thomas explains:


*I couldn’t go to school really because I was an abandoned child…*
[Thomas]

Finishing his sentence, Marie added:


*You were bullied in that way, weren’t you?*
[Marie]

Emily’s story illustrates an example of poor support from the teachers. She explains that she was labelled as a naughty child at school, was regularly suspended from school, and consequently had poor educational attainment:


*Obviously, I wasn’t diagnosed with ADHD till I was like 13, so like in school they used to say that’s just a naughty child. … So it was like always getting suspended, excluded and all that sort of stuff. And in the end [I] went to college and the same happened there.*
[Emily]

These excerpts clearly construct education as a foundation for their low social status as homeless people. All suggest that had they had a stronger educational foundation, participants’ life outcomes would have been different.

#### 4.2.2. Employment

Participants constructed employment as being essentially enabling, providing an appropriate foundation to enable a person to achieve other essential social goods such as health and protection from maladaptive behaviour. Conversely, in this section, participants explain how changes in their job situations set their pathway towards homelessness and illustrate how financial insecurities can lead to maladaptive behaviour. They also constructed employment as enabling them to deal with their underlying mental health problems. They consistently claimed that everything changed when their job situation changed, implying that employment was a foundation on which their lives were anchored. When this foundation was shaken, everything came tumbling down. Alvin, one of the participants who left school at sixteen and built his world around construction work, gives an account of how the change in his job situation set off a downward spiral into homelessness:


*I’ve had it all; really, I’ve had a wife, I’ve got two gorgeous girls now who are 11 and 13, who’ve I got this afternoon actually. Then there was a break up of my marriage. Basically, it started off I was a bricklayer …, I was a builder and had a mortgage had the BMWs had everything… when the recession hit when Liverpool 2008 … won culture capital, it was an abundance of bricklayers so the prices went down in the bricklaying so basically with me having two young children I was the only breadwinner in the family… I had to kinda look for factory work and so I managed to get a job, … it shifted work like four 12 hour days and four 12 hour nights and 6 days off, really hard shifts*



*… I was off Monday, Tuesday, Wednesday, Thursday you know for instance, but I’d treat that like me weekend you know because I’ve worked all weekend. Then… so I’d have a drink then and stuff like that you know. and it’s … 7 o’clock on a Monday morning not really the time to be drinking but I used to treat it like my weekend.*
[Alvin]

According to this participant, losing employment led to the manifestation of several maladaptive behaviours that accelerated his journey along the path to homelessness. These included excessive use of alcohol, leading to strain and ultimately a breakdown in the family relationship. The breakdown exacerbated depression and anxiety, which led to further drinking:


*so we argued me and my ex-missus a little bit and in the end, we split up…, I was diagnosed with depression and anxiety… so I used to drink …to get rid of the anxiety and to numb the pain of the breakup of my marriage really you know it was not good you know.*
[Alvin]

Similarly, Gary identified alcohol as the main issue. However, when one listens to the full story, it transpires that the main issue was the change in his job situation, which led to financial insecurities, excessive drinking, relationship breakdown, and homelessness. Gary explains: 


*I had an injury to my ankle which stopped me from working. I was at home all day, every day. …I was drinking because I was bored. I started drinking a lot [be]cause I couldn’t move about in the house. It was a bad injury I had to my ankle. Um, and one day me and my partner were having this argument and I turned around and saw my little boy just stood there stiff as a board just staring, looking at us. And from that day on I just said to my a partner that I’ll move out, cause I didn’t want my little boy to be seeing this all the time.*
[Gary]

Gary and Alvin’s stories resonate with those of many participants in this study, whereby changes in their employment situation and financial insecurities create conditions that promote alcohol dependency and thus, the breakdown of relationships.

Analysis of their stories reveals that, in all cases, participants constructed employment as providing a protective element against substance dependency. The change in their job situation meant that instead of drinking socially, they drank to cope with financial insecurities and relationship breakdowns. Gary explains:


*I used to have a bit of a drink but it wasn’t a problem because I used to get up in the morning and go out to work and enjoy a couple of beers every evening after a day’s work. Um, but then when I wasn’t working I was drinking, and it just snowballed out, you know snowball effect having four cans every evening and then it went from there. I was drinking more [be]cause I was depressed. I was very active before and then I became like non-active, not being able to do anything and in a lot of pain as well.*
[Gary]

The analysis reveals that participants saw a very strong connection between self and their jobs. For example, they continued to see themselves as a roofer, builder, joiner, etc., even though they no longer worked in those capacities. Danny explains: 


*Yes, yes I was roofing for about 10 years. It was a very good job. I enjoyed what I was doing you know. … [I drank] but not very heavy, just like a sociable drink after work. I’d call into like the local pub and have a few pints and it was controlled. My drinking habit was under control then.*



*I was drinking quite heavily then [when I lost my job and marriage broke down]. I suppose it was a form of release ….*
[Danny]

#### 4.2.3. Health

All participants in this study reported health problems, perceiving good health as an essential step towards getting out of homelessness. Figure 2 illustrates an inextricable connection between health, employment, and homelessness. The analysis reveals that while participants viewed employment as the main route out of homelessness, some were unable to participate in employment due to poor health. 

The data also revealed a connection between health and adopting maladaptive behaviours. All participants reported that depression, anxiety, and stress increased during the course of homelessness; consequently, they increased the use of substances in order to cope with poor health.

#### 4.2.4. Physical Illnesses

Apart from Barry, who sustained injuries before becoming homeless, most participants’ health deteriorated during the course of homelessness. Injuries, drugs, and alcohol-related conditions, such as liver disease, were the most frequently reported physical illnesses. Gary’s story is a typical example of the nature of physical injuries sustained by the number of participants:


*I got assaulted, kicked down a flight of stairs, I landed on my back on the bottom of the stairs but my heel hit the stairs as it was still going up if you know what I mean. Smashed me heel, fractured my heel… So, by the time I got to the hospital and they x-rayed it, there wasn’t even able to operate [be]cause it was in that many pieces, they weren’t even able to pin it if you know what I mean.*
[Gary]

Likewise, Danny’s story provides a typical example of alcohol-related injuries frequently reported by this population group:


*It was an alcohol withdrawal seizure I had at the top of the stairs of all places. So I had a fall and I’ve crushed four of my vertebrae so I’m unable to work at the moment. I am awaiting surgery. Hopefully, that would give me a little bit of mobility back.*



*I’m unable to work at the moment due to my collapse spine. A collapsed spine they put me as a condition. I’ve crushed four vertebrae through a fall.*
[Danny]

Furthermore, the analysis revealed that the unsanitary conditions that homeless people resorted to made them vulnerable to infectious disease, as in the case of Clarke. He explains that he had to sleep “under any shelter I can find where it is not wet”. As a result, he contracted:


*… a foot infection when I was in the streets but that’s gone …my foot was like red, totally red on the bottom and it had all blisters on the bottom. So he [the doctor] gave me the tablets and the cream prescription. I think four tablets a day and the cream twice a day. It seemed to work I suppose. … I was in that much pain and I had to keep sitting down every like 10–15 yards ’cause I was in that much pain. I keep getting sent home from work ’cause I was in that much pain.*
[Clarke]

In all cases, participants perceived their poor health to be a setback in their ambition to find employment and get themselves out of homelessness. As Clarke explains:


*… I was quite scared I suppose ’cause I didn’t know where to go or what was gonna happen with my job ’cause I was living on the streets.*
[Clarke]

Similarly, the desire to regain employment is one of Alvin’s motivations to improve his health. He explains:


*I’m striving towards you know …I wanna make a success of my life, I wanna rebuild and get that BMW back and get you to know.*
[Alvin]

Alvin’s story resonates with those of many participants in this study: they constructed health as a necessary foundation to build on in order to achieve other goals, such as employment and getting out of homelessness.

#### 4.2.5. Mental Wellbeing

All participants in this study reported anxiety, depression, and stress. Of those who had pre-existing mental health problems, their symptoms deteriorated during the course of their homelessness, and they consequently increased the use of drugs and alcohol to cope.

Alvin, Clarke, and Gary’s stories made a strong connection between mental health issues, changes in employment situations, and the use of drugs and alcohol as a maladaptive coping strategy. This was evident in Alvin’s repeated references to “drinking to numb the pain”: 


*… so we argued me and my ex-missus a little bit and in the end, we split up so…, the drinking got worse, I was diagnosed with depression and anxiety, now I used to drink used to drink to get rid of the anxiety and also to numb the pain of the breakup of my marriage really you know it wasn’t good you know, one thing led to another and I just couldn’t stop it got hold of me the alcohol.*
[Alvin]

Gary puts it:


*They say… well I suffer really bad now from mental health problem and anxiety and they say like at the age of 11, to lose a parent at the age of 11 you like in between. I was drinking more cause I was depressed.*
[Gary]

Clarke adds:


*I was drinking quite a lot as well trying to take my to mind off it and trying to make me go to sleep. ’Cause I was up all night cause I couldn’t get to sleep. But if I drank had a few drinks I could go sleep.*
[Clarke]

In the worst times, participants reported that they had suicidal thoughts, claiming that it was painful thoughts of ending their lives that led them into alcohol and substance dependency: 


*I was admitted about I don’t know 10 times—suicidal thoughts, pancreatitis which was through the drinking, just hating myself. …, did not really wanna be on this planet. The reason I used to drink the vodka is that it sooner is asleep than awake. I used to hate waking up cause I just hated what was going on. Hated what I was doing to my family but it was a catch 22 cause I drank more to block it out, you know.*
[Alvin]

As another participant puts it: 


*There have been days that I feel like I just don’t want to be here anymore, I don’t want to face the world; and it’s very frightening, very frightening.*
[Danny]

Henry explains:


*I’m going to end up waking up and thinking of killing meself like today. It’s wrong. I shouldn’t have to feel like that. I’m 29, and all I want to do really is just die.*
[Henry]

Danny describes an internal struggle to suppress a strong desire to take substances to numb the pain of low mood, and his fear of reverting to substance dependency:


*I suppose, your mood’s very low when you’re homeless and you’re not able to think straight and you would possibly turn to anything you know. Drink or drugs. At the moment, like I said I’m being strong, I’m not turning back to alcohol as I’d like to think I’d never go back down that path again.*



*It’s my mental health that suffered a lot I was very, very depressed. Um, which I did. I have tried to take my own life a couple of times through medication and self-harm cause I couldn’t put up any longer on the streets you know.*



*I think if I was in now I’d probably have tried to take my life again. Not being able to handle or cope with not having the help and support.*
[Danny]

The recurrent phrase was “when you homeless you can’t think straight”. Many participants shared this view, suggesting that a consequence of not being able to think straight is that you end up taking substances. For example, Lee stated:


*When you homeless you can’t think, you can’t do anything, the best thing people do is to get pissed get drunk and drugs just to get by, when they get drunk they don’t bother where they go. I don’t drink alcohol but I have used cannabis in the past.*
[Lee]

## 5. Discussion

A key feature that distinguishes this study from comparable previous studies is that it employed the constructivist grounded theory to develop a model to examine the potential pathways by which socioeconomic and political conditions, individual circumstances, and maladaptive behaviour cause homelessness. We also explicitly acknowledge that the methodology we employed means that our resulting theoretical explanation constitutes our interpretation of the reasons and the contexts that participants described that led to them to becoming homeless. Figure 2 provides a visual illustration of the interaction of factors that explain the pathways to homelessness.

The central concept in this study, “being at the bottom rung in an unequal society”, captures participants’ descriptions of their perceived social status. We interpreted “being at the bottom rung in an unequal society” as participants’ attribution of their homelessness status to an uneven distribution of social goods, such as education, employment, and health, that are known to be the fundamental determinants of unequal social status. Several public health and social justice theorists proposed that the unequal distribution of these social goods is a matter of social justice [42,43,44,45,46]. The strength of examining the data with a district philosophical lens is that it provides congruence between the research question and researchers’ epistemological and ontological assumptions. The limitation is that it has the potential to narrow researchers’ view of the data. 

Participants ascribe their homelessness to poor outcomes in education, employment, and health. The English Indices of Multiple Deprivation (IMD) [47] identifies these as three of the seven domains of deprivation. These findings are consistent with several studies that show that a cluster of disadvantages, including poverty, poor education, and poor employment or unemployment, account for most causes of homelessness [6,16,17,25,48]. For example, the study by Watson et al. [25] found extreme levels of poverty and social exclusion amongst homeless people. Therefore, if these three domains of deprivation—education, employment, and health—are the main causes of homelessness, it follows that an upstream approach to tackling homelessness should focus on tackling deprivation.

Participants’ description of education revealed that they perceive education as essentially enabling. Participants’ stories imply that they reject the two-tier education system: that is, education for the poor and education for the affluent. They propose an education that enables social mobility and asserts that for education to facilitate social mobility, it has to occur in early childhood. These findings are consistent with a classic study by Bassuk et al. [49], which reported that graduating from high school and having a larger social network has a protective effect against homelessness. Similarly, a study by Tierney, Gupton, and Hallett [50] indicated that education plays a critical role in how adolescents mature into adults. 

Participants’ descriptions of their educational experiences appear to imply that their adverse childhood experiences (ACEs) contributed to their poor education outcome, as well as restricted their social mobility. Table 1 shows that fifteen participants in this study experienced more than one ACEs, including abuse/neglect and growing up in children’s institutions or being raised by single parents. These findings are consistent with previous studies that show adverse childhood experiences tend to cluster together, and that the number of adverse experiences may be more predictive of negative adult outcomes than particular categories of events [22,27,48,51]. Previous studies have shown that a cluster of childhood problems, including mental health and behavioural disorders, poor school performance, a history of foster care, and disrupted family structure, was most associated with adult criminal activities, adult substance use, unemployment, and subsequent homelessness [22,27,52].

Figure 2 indicates that participants posited that education should enable a young person to develop positive social connections and social networks and harness skills in social interaction. It has been previously reported that these skills are essential in enabling homeless people to successfully engage with social institutions, such as education, health, and social services, and thus take advantage of available opportunities to improve their social status [6]. Similarly, participants in the study by Tierney et al. [50] reported that a lack of a positive social network was one of the main factors that influenced their educational experience and their homelessness status. This view of education as a vehicle for social mobility has been recognised by the UK Department of Education (DE) in its publication Unlocking Talent, Fulfilling Potential: A Plan for Improving Social Mobility through Education, in which the UK government pledged to focus on the role of education in improving social mobility [53]. It expresses an ambition to increase higher education opportunities for young people from the most disadvantaged backgrounds [53]. However, it is unclear how the kind of education that the DoE proposes will enable positive social connections, networks, and interactions at an age proposed by participants in this study.

Participants constructed employment as a foundation on which their lives were secured. Similarly, the Caton et al. [54] study constructed employment and earning an income as having a preventive effect against homelessness. Participants in the current study and in other studies reported that losing a job, inability to pay rent, and a general lack of funds were the major contributors to becoming homeless, and indicated that insecure job markets adversely affect individuals in low-paying jobs who are particularly vulnerable to becoming homeless [55]. It is worth noting that in welfare societies, poor people have a secure income; however, participants’ stories indicated that income without a job does not provide a sufficient protective mechanism against homelessness. The two have two distinct functions—the job provides a sense of purpose, and the income provides means to acquire basic essentials for living.

Participants reported an inextricable connection between their health and homelessness. Likewise, several studies reported an association between homelessness and higher rates of infectious disease, injury, substance use, psychiatric illness, and accident and emergency admissions [56,57,58,59,60]. 

While this connection has been widely reported, the difference between the current and other studies is the way this connection is being made. 

Participants appear to describe the link between homelessness and health in the same way as Seedhouse’s [61] theory of “health foundations for achievement”, in which he posited that membership of social class is not only an obstacle to accessing social goods but can also create further obstacles, such as lifestyle determined disease and illness [61]. Several studies demonstrated the inextricable connection between the membership social group, health, and homelessness. For example, Bingham et al. indicated that in Canada, being of an indigenous ethnicity independently predicted the likelihood of being homeless at an early age [7]. Similarly, the studies by Bauermeister et al. and Skosireva et al. indicated that discrimination against LGBTQ2S and those with mental illnesses or cognitive disabilities increase their vulnerability to homelessness [13,14,15]. In turn, the marginalisation they experience prevents them from accessing other social goods, such as education, social services, and employment. 

## 6. Conclusions

This study set out to examine the stories of homeless people in order to document what they ascribe to their social status and to propose a conceptual explanation. It revealed that participants conceptualised being homeless as “being at the bottom rung of the ladder in an unequal society”. They catalogued their adverse childhood experiences, which they believe reduced their capacity to fully engage with the social institution for social goods, such as education, social services, and institutions of employment. Since not all people who have misfortunes of poor education, poor health, and loss of job end up being homeless, we contend that a combination of these with multiple adverse childhood experiences identified by a participant in this study may have weakened their resilience to traumatic life changes, such as loss of job and poor health. We, therefore, propose that policymakers and providers of the services for homelessness people should devote equal attention to tackling the fundamental determinants of homelessness as is granted in dealing with behavioural causes.

## Figures and Tables

**Figure 1 ijerph-16-04620-f001:**
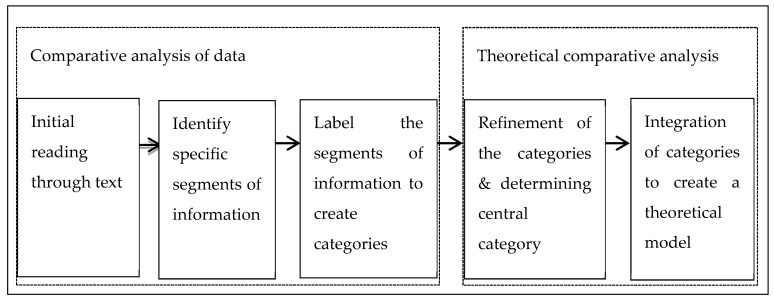
Summary of the process of data analysis and theory building (Mabhala, 2013 [30]).

**Figure 2 ijerph-16-04620-f002:**
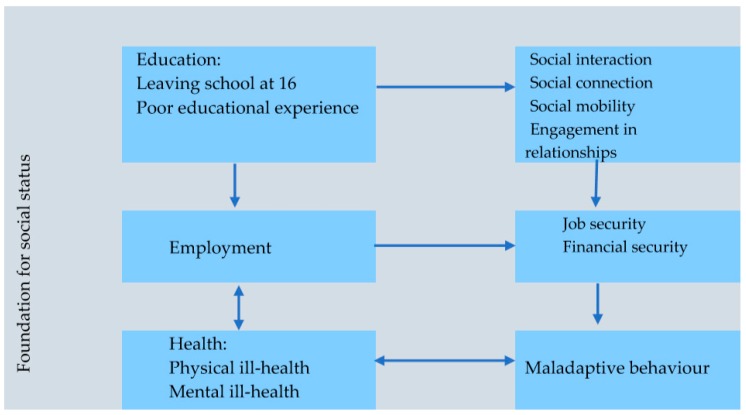
Illustrates participants’ perceived role of education, employment, and health in shaping the conditions that led to their low social status.

**Table 1 ijerph-16-04620-t001:** Participants’ demographic information.

Pseudonym	Age	Education History	Childhood Living Arrangements	Employment History	Drugs, Alcohol or Tobacco?	Age of Contact with the Criminal Justice System
Ruddle	35	Left school at 16	Both parents	Never worked	Yes	18
Ian	74	Both parents	Both parents	Paint and decoration	Yes	No criminal record
Patrick	64	Left school at 16	Both parents	Never worked	Yes	12
Lee	35	Left school at 15	Children’s care	Chef	Yes	15
David	31	Left school at 16	Foster care	Untrained chef	Yes	No criminal record
Marco	46	Left school at 16	Residential school for boys with challenging behaviour	Building construction	Yes	15
Alvin	39	Left school at 16	Both parents	Builder	Yes	No criminal record
Barry	43	Law Degree	Both parents	Military	No	No criminal record
Clarke	18	Left school at 16	Mum and stepdad	McDonald	Yes	No criminal record
Danny	44	Left school at 16	Both parents	Roofer	Yes	No criminal record
Emily	23	Left school at 16	Single mum	Morrison supermarket	Yes	16
Finn	24	Left school at 16	Single parent	Takeaway restaurant	Yes	No criminal record
Geoff	32	Left school at 16	Both parents	Warehouse	Yes	No criminal record
Gary (John)	35	Left school at 16	Mum and stepdad	Joiner	Yes	No criminal record
Guy	26	Left school at 16	Both parents	Factory worker	Yes	No criminal record
Henry	29	Left school at 15	Stepdad and mum	Electrician	Yes	16
Ian Cath	42	Left school at 16	Both parents	Factory worker	Yes	16
Tom	36	Left school at 13	Children’s home	Never worked	Yes	No criminal record
Marie	35	Left school at 13	Children’s care home	Never worked	Yes	No criminal record
Leo	42	Left school at 16	Single mum	Plumber, computer tech and supermarket	Yes	No criminal record
Norma	31	Left school at 16	Foster care	Cleaner	Yes	No criminal record
Crewe	58	Left school at 15	Foster care	Motor mechanic	Yes	16
Paddy	35	Left school at 16	Mum and stepdad	Never worked	Yes	14
Matt	33	Left school at 16	Foster care	Builder	Yes	18
Milly	31	Left school at 16	Foster care	Never worked	Yes	16
Kieran	38	Left school at 16	Children’s home	Motor mechanic	Yes	18

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
