# Peer review of "Being at the Bottom Rung of the Ladder in an Unequal Society: A Qualitative Analysis of Stories of People without a Home"

_ijerph, 2019, doi:10.3390/ijerph16234620_

Round 1
Reviewer 1 Report
The paper needs careful proof reading and would benefit by attention to English language style. The methods sections needs to be tighter and more clearly articulated with more detail on accessing participants (who and where di this take place), consent, confidentially, anonymity. There needs to be an indication of how the full range of interviewees have been quoted (represented) as opposed to a select few. Otherwise a good paper, which should be of interest to a wide readership
Author Response
Being at the bottom rung of the ladder in an unequal society: A qualitative analysis of stories of people without a home
Dear Reviewer,
We are grateful for taking your time making sure that our paper is as good as it can be.
We believe that your feedback and advice has helped to strengthen this paper. We continue to believe that our work is best placed in your journal.
We have considered your comments and attempted to respond to them systematically. We created two copies marked and clean copies of the revised article. However, realise that the number of revisions that we made makes it difficult to read the marked copy, so we sent you unmarked. In our response we indicated precisely where we made changes.
Please let me know if you prefer a marked copy.
Yours sincerely
Reviewer 1
Comment: The paper needs careful proofreading and would benefit by attention to English language style.The methods sections need to be tighter and more clearly articulated with more detail on accessing participants (who and where did this take place), consent, confidentially, anonymity.
Response: Thank you for your comments. We have edited section 2.1 “setting, recruitment and sampling strategies” lines 99 – 105 describe the study setting, explain the criteria for selecting setting, specify the underpinning principle for sites and participants’ selection.
Lines 106 – 109 explain how the study was introduced to the participants to facilitate informed consent.
Lines 110 – 1118 explain the criteria used to help potential participants make a self-assessment of their eligibility to participate without unfairly depriving others of the opportunity; participants information sheet outline criteria that potential participants had to meet.
We also edited section 2.3 “Ethical approval”, lines 150 - 156 explain how confidentiality and anonymity were preserved.
Lines 157 to 159 explain how the informed consent was obtained.
Comment: There needs to be an indication of how the full range of interviewees have been quoted (represented) as opposed to a select few.Response: We edited section 3 “data analysis” line 215 – 216 explains how the quotes were selected.
Comment: Otherwise a good paper, which should be of interest to a wide readershipResponse: We found your comments valuable in further improving the quality of our paper. We hope we address them to your satisfaction. As you will understand we had to ensure a balance in addressing your comments along with other three reviewers and maintain the shape and flow of the article.

Reviewer 2 Report
Much research has been done on homelessness in the United States, Canada, and FEANTSA. Authors such as Gaetz et al, could have assisted with the context of homelessness for this paper, which I think is thin. I am always leery of a number of self-citations, even as they may relate to the topic at hand. Overall, I think inclusion of a greater range of views and sources would assist here.
I would also like to see more exploration of the findings in relation to Seedhouse's work on health foundations for achievement. A more nuanced exploration of the relationships between health, work, and resilience would be very interesting. In that exploration I think may lie something that could be a real contribution to the field. I suspect your analysis may have more to offer than is shown here.
Author Response
Being at the bottom rung of the ladder in an unequal society: A qualitative analysis of stories of people without a home
Reviewer 2
Dear Reviewer,
We are grateful for taking your time making sure that our paper is as good as it can be.
We believe that your feedback and advice has helped to strengthen this paper. We continue to believe that our work is best placed in your journal.
We have considered your comments and attempted to respond to them systematically. We created two copies marked and clean copies of the revised article. However, realise that the number of revisions that we made makes it difficult to read the marked copy, so we sent you unmarked. In our response we indicated precisely where we made changes.
Please let me know if you prefer a marked copy.
Yours sincerely
Comment: Much research has been done on homelessness in the United States, Canada, and FEANTSA. Authors such as [1]could have assisted with the context of homelessness for this paper, which I think is thin. I am always leery of a number of self-citations, even as they may relate to the topic at hand. Overall, I think inclusion of a greater range of views and sources would assist here.
Response: Thank you for your observation. We felt that rather than adding a few references to the introduction section we would take a comprehensive revision of the whole section thus, maintain the flow of argument. Lines 42 down to including line 92 are have been revised.
Comment: I would also like to see more exploration of the findings in relation to Seedhouse's work on health foundations for achievement. A more nuanced exploration of the relationships between health, work, and resilience would be very interesting. In that exploration I think may lie something that could be a real contribution to the field. I suspect your analysis may have more to offer than is shown here.Response: Thank you, for your interest in work of my colleague, David Seedhouse. We have expanded as much as we can within the limit of this article (see lines 627 – 637). Admittedly, it was difficult to engage fully in David’s philosophical discussion while maintaining the shape of this article.
Once again thank you for your time in reviewing this article, your comments served to strengthen it.

Reviewer 3 Report
In this article the authors aimed to examine the stories of homeless people, in order to document their perceptions of their social status, the reasons that led to their homelessness and propose a conceptual explanation.
This article is well designed, and written. The use of qualitative study, with grounded theory, is relevant. It led to consistent results and implications.
I propose mostly revision on the form and the organization of sections (cf revisions proposed on chapter methods).
Please find below my specific comments and suggestions about this article :
Summary
It would be usefull to describe there, in section methods, which approach was chosen by authors for qualitative analyses (grounded theory ).
Introduction :
The introduction brings well the study.
Before the aim, maybe add : « IN this study… » or write « this study aimed… »
The objective are clear and specific.
I’m surprised about the lack of studies concerning homeless people’s perception on factors which led them to hemolessness. I don’t have specific study to propose to the authors, but i would like to ask them if they are really sure that there is none ? For exemple, there is a lot of works which permitted to understand to precariousness/vulnerability process (Larcher et al., Castel et al. , …). These works used, at less in part, qualitative studies.
Methods :
The choice of CGT is correclty argued and appears to be relevant.
The sampling strategie is also very well described, and relevant for CGT. Why did only 4 wmoen could be included ?
I agree with the term ‘theoretical sufficiency’.
Process fr data analysis is also well described. You cold also refer to Pailllé, to describe the grounded theory analysis process.
On line 24, the paragraph « All participants attributed being homeless to their adverse childhood experiences (ACEs)… » should be placed in the results sections.
The figure 2, and the synthesis of main categories which have emerged has also to be placed in the results (or discussion) section.
It would be usefull to propose the COREQ criteriae as attached document.
Results :
The results are also relevant and well described.
About the role of education, i’m not sure to understand : in Pat’s experience or view, education did not have influenced his condition, is it right (when you write « Pat implies that had he been educated he would not have been in a better position in society » ) ? If it’s that, i think that it is not relevant to speak about education as a perceived factor which led to homelessness, on this part of article (or brin git with the arguments and veiws of the other that are described after). Indeed, even if they all but one had left school before 16 or had poor education, we can’t have consistent quantitative results on 26 persons. But their reality is more that for them education did not play a rôle. So maybe it could be intersting to discuss then why this contrast. (but maybe i did not well understand the sens of this section). For Pat, as explained in the discussion section, it could traduced that he rejected the education system ? Maybe clarify this section.
After this paragraph, the perceived influence of end of shcool is however well described.
Discussion :
I did not see a section speaking about weaknesses of the study ? for exampl, why only ‘ women.
Conclusion :
Line 458 : the --> this, or that ?
With my best regards

Author Response
Reviewer 3
Comment: In this article, the authors aimed to examine the stories of homeless people, in order to document their perceptions of their social status, the reasons that led to their homelessness and propose a conceptual explanation.
This article is well designed and written. The use of qualitative study, with grounded theory, is relevant. It led to consistent results and implications.
Response: Thank you very much; we appreciate your observation.
Comment: I propose mostly revision on the form and the organization of sections (cf revisions proposed on chapter methods).
Response: Thank you we have take your recommendation along with other review and tried to strike a right balance between all reviewers recommendations.
Comment: Please find below my specific comments and suggestions about this article:
- Summary
Comment: It would be useful to describe there, in section methods, which approach was chosen by authors for qualitative analyses (grounded theory ).
- Introduction :
Comment: The introduction brings well the study.
Response: Thank you, we have extensively revised the introduction in response to the other reviewer. Hopefully, it reads even better now. Introduction lines 42 – 93 have been revised
Comment: Before the aim, maybe add : « IN this study… » or write « this study aimed… »
Response: Thank you we have revised this sentence inserted your suggestion (see line 92)
Comment: The objective are clear and specific.
Response: Thank you, we appreciate that
Comment: I’m surprised about the lack of studies concerning homeless people’s perception on factors which led them to hemolessness. I don’t have specific study to propose to the authors, but I would like to ask them if they are really sure that there is none? For example, there is a lot of works which permitted to understand to precariousness/vulnerability process (Larcher et al., Castel et al. , …). These works used, at less in part, qualitative studies.
Response: Thank you the other reviewer proposed that we add more. Studies which we did in a revised version hopefully the revised version addresses your concern see lines 42 -93).
- Methods:
Comment: The choice of CGT is correctly argued and appears to be relevant.
Response: Thank you.
Comment: The sampling strategy is also very well described, and relevant for CGT. Why did only 4 women could be included ?
Response: We have revised the methodology, included the explanation on why there was a gender imbalance. The parts of the methods that we revised are: line 143 to 153, lines 181- 190
Comment: I agree with the term ‘theoretical sufficiency’.
Response: Thank you
Comment: Process for data analysis is also well described. You could also refer to Pailllé, to describe the grounded theory analysis process.
Response: Thank you for your suggestions, we could not access the full text of Pailllé, within the time editor gave us to turn this revision (five days)
Comment: On line 24, the paragraph « All participants attributed being homeless to their adverse childhood experiences (ACEs)… » should be placed in the results sections.
Response: Thank you we made alteration you suggested
Comment: The figure 2, and the synthesis of main categories which have emerged has also to be placed in the results (or discussion) section.
Response:Thank you for your suggestion we have moved figure 2 to the discussion ` section
- Results :
Comment: The results are also relevant and well described.
Response: thank you very much, we appreciate that
Comment: About the role of education, I’m not sure to understand: in Pat’s experience or view, education did not have influenced his condition, is it right (when you write « Pat implies that had he been educated he would not have been in a better position in society » ) ? If it’s that, I think that it is not relevant to speak about education as a perceived factor which led to homelessness, on this part of article (or brin git with the arguments and veiws of the other that are described after). Indeed, even if they all but one had left school before 16 or had poor education, we can’t have consistent quantitative results on 26 persons. But their reality is more that for them education did not play a rôle. So maybe it could be intersting to discuss then why this contrast. (but maybe i did not well understand the sens of this section). For Pat, as explained in the discussion section, it could traduced that he rejected the education system ? Maybe clarify this section.
Response: Thank you, we felt that Pat’s story of incompletion of education is a significant incident in homelessness and was well supported by other studies. If that is okay with you we would be grateful to leave this as it is.
Comment: After this paragraph, the perceived influence of end of school is however well described.
Response: Thank you very much.
Discussion:
Comment: I did not see a section speaking about weaknesses of the study? for example, why only ‘women.
Response: Thank you for your observation. The weaknesses of this study were integrated as part of the discussion. See line 570
